# Anti-Cancer and Medicinal Potentials of Moringa Isothiocyanate

**DOI:** 10.3390/molecules26247512

**Published:** 2021-12-11

**Authors:** Yu-Yao Wu, Yan-Ming Xu, Andy T. Y. Lau

**Affiliations:** Laboratory of Cancer Biology and Epigenetics, Department of Cell Biology and Genetics, Shantou University Medical College, Shantou 515041, China; 21yywu1@stu.edu.cn (Y.-Y.W.); amyymxu@stu.edu.cn (Y.-M.X.)

**Keywords:** *M. oleifera*, anti-cancer, isothiocyanates, MIC-1

## Abstract

*Moringa oleifera* (*M. oleifera*), which belongs to the Moringaceae family, is a common herb, rich in plant compounds. It has a variety of bioactive compounds that can act as antioxidants, antibiotics, anti-inflammatory and anti-cancer agents, etc., which can be obtained in different body parts of *M. oleifera*. Isothiocyanates (ITCs) from *M. oleifera* are one class of these active substances that can inhibit cancer proliferation and promote cancer cell apoptosis through multiple signaling pathways, thus curbing cancer migration and metastasis, at the same time they have little adverse effect on normal cells. There are multiple variants of ITCs in *M. oleifera*, but the predominant phytochemical is 4-(α-L-rhamnosyloxy)benzyl isothiocyanate, also known as moringa isothiocyanate (MIC-1). Studies have shown that MIC-1 has the possibility to be used clinically for the treatment of diabetes, neurologic diseases, obesity, ulcerative colitis, and several cancer types. In this review, we focus on the molecular mechanisms underlying the anti-cancer and anti-chronic disease effects of MIC-1, current trends, and future direction of MIC-1 based treatment strategies. This review combines the relevant literature of the past 10 years, in order to provide more comprehensive information of MIC-1 and to fully exploit its potentiality in the clinical settings.

## 1. Introduction

According to the latest survey released by the International Agency for Research on Cancer of the World Health Organization, the number of cancer deaths worldwide is rising [1], and the methods commonly used for cancer treatment are percutaneous ablation treatments, surgical resection, and chemotherapy [2,3,4]. Although these methods effectively improve the survival rate in cancer patients, the overall survival rate is still very low. The lack of effective treatment programs cause serious health problems that are necessary and urgent for early cancer patients. At present, medicinal plants have entered the vision of cancer researchers, which become an important source for drug development. Screening, extracting, and exploring new compounds from Chinese herbal medicine is an important way to develop efficient and low-toxicity anti-cancer drugs [5,6,7].

*Moringa oleifera* (*M. oleifera*) is a perennial tree, 3–13 m tall, native to India, due to its introduction to some countries, is widely planted in tropical areas, also cultivated in Taiwan and Guangdong of China (Figure 1) [8]. It is a tropical multifunctional tree with all parts of medical value for a variety of diseases, including fever, asthma, viral infection, cancer, etc. [9]. *M. oleifera* seeds are a very promising resource, due to their content of monounsaturated fatty acids and important bioactive compounds including alkaloids, glucosinolates (GLs), isothiocyanates (ITCs) and thiocarbamates [10]. They are commonly used in folk medicine to treat stomach pain, ulcers, poor vision, joint pain, and help digestion [9]. Studies show that the *M. oleifera* seed ingredients have the coagulant, bactericidal, fungicidal, and insecticidal properties [11] and administrating *M. oleifera* seeds powder can significantly protect animals from oxidative stress and reduce tissue arsenic concentration [12]. The leaves are the most used part of the plant, and the leaves of *M. oleifera* are rich in vitamins, carotenoids, polyphenols, phenolic acids, flavonoids, alkaloids, glucosine, isothiocyanate, tannins, and saponins [13]. Studies have demonstrated that *M. oleifera* leaves are beneficial for several chronic diseases, including hypercholesterolemia, hypertension, diabetes, insulin resistance, non-alcoholic liver disease, cancer, and overall inflammation [14]. While in the bark of the roots and stems, the protocyanogen protein is contained. Extracts from *M. oleifera* leaves, roots, stem, and bark all exhibit strong antioxidant activity in vitro [15]. *M. oleifera* flowers are high in protein, dietary fiber, and total phenols [16]. All of these make *M. oleifera* exhibiting properties including anti-proliferation, liver protection, anti-inflammation, anti-atherosclerosis, oxidative DNA damage protection, anti-peroxidation, and heart protection [17]. Current studies have shown the potential of *M. oleifera* as an anti-cancer drug, and research has found that its extract can inhibit cell proliferation, growth, and migration by affecting different signaling pathways in cells [18,19]. Similar to selenium (Se) in cancer prevention, *M. oleifera* extract is also able to promote cancer apoptosis through mitochondria, such as studies showing that *M. oleifera* extract has significant anti-cancer effects on melanoma cells in vitro involving mitochondria-mediated caspase enzyme-dependent and caspase enzyme-independent apoptotic pathways [20,21]. At the same time, *M. oleifera* has a great potential to inhibit tumor progression without affecting the normal physiology and function of the body, and therefore can be used as a cancer treatment drug.

ITCs are natural small molecules formed by glucosinolate precursors in cruciferous vegetables, which reduce the activation of carcinogens and increase the detoxification of carcinogens [22]. The ITCs formed by the biotransformation of *M. oleifera* GLs contain additional sugars in their chemical structures, which provide stability for these bioactive compounds compared to other ITCs found in other crops [23]. MIC-1 is one of ITCs, especially enriched in the seeds of the *M. oleifera* [24], and is so far the predominant phytochemical in *M. oleifera* seed extraction [25,26]. One of its most remarkable biological properties is its anti-inflammatory role [27]. Inflammation is a protective response of the body to a variety of internal or external stimuli derived from physical, chemical, or biological sources, including physical forces, radiation, extreme temperature, stimuli, pathogens, and metabolic overload, and increasing evidence suggests that chronic inflammation leads to cancer [28,29]. During chronic inflammation, activated macrophages are thought to cause continuously deteriorating inflammation through sustained and excessive production of inflammatory mediators such as cytokines, chemokines, lysozymes, proteases, growth factors, and eicosanes [30]. MIC-1 can cause the inability of activation of several transcription factors to inhibit the occurrence and development of inflammation [26]. This shows the anti-cancer potential of MIC-1.

This paper provides updates on MIC-1 from *M. oleifera*, combined the relevant literature for nearly 10 years, mainly from studying its anti-cancer property, in order to provide ideas for future tumor research and treatment.

## 2. Research Progress in *M. oleifera*

### 2.1. Bioactive Compounds from M. oleifera

Moringa is a valuable medicinal plant, different parts contain a variety of bioactive substances, and all parts of *M. oleifera* have medical value and can be overlapped to treat a variety of diseases [9]. Different parts of *M. oleifera* also contain different amounts of bioactive substances, carbohydrates is the most abundant in the roots and the least in the leaves [31]. Also the main material content varied with different treatments, with chlorogenic and gallic acids being the predominant phenolic acids in the raw leaf powder, while caffeic and gallic acids were higher in pretreated leaf powder [32]. Prabakaran et al. [33] determined five sites of *M. oleifera* by five different extraction methods and found that phenolics, flavonoids, myricetin, and quercetin had the highest content in leaves and gentisic acid and biochanin A were the highest in roots. At the same time, they also found that extraction of *M. oleifera* with ethyl acetate, ethanol, and methanol can obtain the maximum amount of potentially beneficial compounds in the extract and may contribute to the treatment of certain diseases. Except for the root, the content of glucomoringin was predominant in all body parts of *M. oleifera* [34]. Table 1 shows the major bioactive substances contained in different parts of *M. oleifera*.

### 2.2. M. oleifera: A Promising Anti-Cancer Agent

In anti-cancer studies, *M. oleifera* were found to slowdown the cancerous process through targeting chemoprevention, inhibiting carcinogen activation and inducing carcinogen detoxification, anti-inflammation, anti-proliferation of tumor cells, and inducing apoptosis of cancer cells [45,46].

While studying *M. oleifera* leaves, the researchers found that *M. oleifera* leaf extract could affect the growth of cancer cells through different signaling pathways. A study on human prostate cancer cell lines (PC-3), it was found that *M. oleifera* leaf extract controls PC-3 growth through three signaling pathways. After grinding the *M. oleifera* leaves into powder, being extracted with ethyl acetate at room temperature and then treating the obtained extract found that the *M. oleifera* extract was capable of down-regulating GLI1 transcription factor and SMO protein mRNA expression in Hedgehog signaling pathway, the signaling pathway plays an important role in cell proliferation and differentiation. Meanwhile, the extract can also induce reactive oxygen species (ROS) production, promoting apoptosis through the activation of caspase 3. There is concentration-dependent effect of cell cycle stagnation in G0/G1 [47]. The research by Luetragoon et al. [48] found that the extract of *M. oleifera* leaf can up-regulate cysteine, downregulate the anti-apoptotic factors, and induce apoptosis of cancer cells. Besides, studies by Tiloke et al. [49] found that *M. oleifera* leaf extract significantly increased lipid peroxidation, DNA damage, and γH2AX levels, while causing significant increases in caspase 9, caspase 3/7, Bax, Smac/diablo, and cleaved PARP-1, inducing apoptosis. Reducing the proliferation of cancer cells is a way to control cancer cell amplification, and *M. oleifera* leaf extract significantly reduces the levels of c-myc, p-Bcl2, and Hsp70 to induce cell cycle stagnation [50]. A study of human non-small-cell lung cancer cell line (A549) by Xie et al. [51] found that the alkaloid extract of *M. oleifera* significantly inhibited p-JAK2 and p-STAT3 and has dose-dependent effects, these results suggest that the *M. oleifera* leaf extract works by inhibiting the JAK2/STAT3 signaling pathway, it is closely related to the proliferation, angiogenesis, invasion, and migration of non-small-cell lung cancer, which induced the apoptosis and cell cycle block of A549 cells. In addition to the above signaling pathway, *M. oleifera* leaves can also affect the activity of cancer cells through other ways, such as apoptosis mediated by the caspase pathway found by Do et al. [52], they use methanol extract, hexane fraction, chloroform fraction, ethyl acetate fraction, and water-soluble fraction to expose human melanoma cells, found that the leaf extract of *M. oleifera* induced melanoma cells to participate in the caspase-dependent and caspase-independent apoptotic pathways mediated by mitochondrial ROS, and promote tumor cell apoptosis.

Abd-Rabou et al. [53] found that *M. oleifera* seed oil can induce apoptosis of Caco-2 and HCT116 cells of colorectal cancer by inducing mitochondrial dysfunction, while unaffecting normal cells. How mitochondria mediated apoptosis still requires further experimental verification. Studies have shown that the water extract obtained from *M. oleifera* is anti-proliferative and promoting apoptosis, but they have no effect on healthy peripheral blood monocytes. Moreover, it was found that the pro-apoptotic effect of water extract in *M. oleifera* seeds was correlated with a decrease in protein expression of B-cell lymphoma 2 (BCL2) and sirtuin-1, which are involved in the apoptosis process [54]. The oil–water extract of *M. oleifera* seeds appears to differentiate between regulating the proliferation and apoptosis of healthy and cancer cells, and this ability may be associated with the microRNA present in the extract [54]. It is also speculated by Potestà et al. [54] that microRNA in extract can differentiate between healthy cells and cancer cells, and regulate their proliferation and apoptosis, respectively. Through further studies, specific human apoptosis-related target genes for plant miRNA were identified by Potestà et al. [55]. They found seed water extracts can lead to a decrease in BCL2 protein expression in tumor cells and a reduced mitochondrial membrane potential, and both are associated with cell apoptosis. For two scientists studying breast cancer, Adebayo et al. [56] ground *M. oleifera* seeds into powder, extracted their plant chemicals with ethanol, and then treated breast cancer cells (MCF7), observing that the extract of *M. oleifera* seeds significantly inhibited the proliferation of the cell line. Elsayed et al. [57] also found seed essential oil from *M. oleifera* has potent cytotoxic activities against MCF7. However, its specific mechanism needs further research and determination.

With relatively limited cancer studies on other parts of *M. oleifera*, Patriota et al. [58] treating sarcoma 180-bearing mice with trypsin inhibitor from *M. oleifera* flower found that it can cause significant tumor weight loss (90.1–97.9%). Al-Asmari et al. [59] found that bark extracts showed significant anti-cancer properties for MDA-MB-231 and HCT-8 cancer cells and had significantly lower cell survival in both cell lines, while the number of apoptotic cells in cancer cell lines increased significantly. Besides, the extract effectively stops the cell cycle in G2/M phase. Vasanth et al. [60] used *M. oleifera* stem bark extract to develop anti-cancerous colloidal silver, allowing the substance to treat HeLa cells and find that it can induce HeLa apoptosis by the production of ROS. The results show that the bark extract of *M. oleifera* has anti-cancer activity and can be used to develop new drugs for breast cancer and colorectal cancer, but the specific mechanism of action needs to be further studied. At the same time Siddiqui et al. [61] found that extract of *M. oleifera* fruit can reduce the proliferation of HepG2 cells while significantly reducing cell viability through ROS-mediated apoptosis and activation of caspase 3 enzyme activity. Abundant natural phytochemical is a potential in cancer treatment and daily nutrients can be a treasure trove of anti-cancer drugs. It is widely accepted that nanotechnology could be a future direction in cancer treatment. The application of phytochemicals in combination with nanotechnology amplifies the therapeutic effect and provides a new way to solve the difficult economic and environmental problems of nanotechnology. Therefore, combining phytochemicals with nanotechnology is a promising approach. However, challenges of nanotechnology are not yet fully settled. One of challenges in drug delivery is the safety for human health, as nanomaterial may not have immediate impact quickly. Furthermore, the manufacturing of nanomedicine products for commercialization is a key obstacle, as large-scale production is technically challenging and their physicochemical properties may vary from batch to batch. Though there has been great progress in the application of nanophytochemicals in the treatment of cancer, we have a long way to large-scale application of nanoparticles for drug delivery and many questions remain.

## 3. The Introduction of Isothiocyanates

### 3.1. Basic Information of Isothiocyanates

ITCs, sulfur-containing bioactive compounds [62] resulting from enzymatic hydrolysis of GLs [63], appear in many Brassicaceae vegetables [64] with a variety of biological functions, including antibacterial [65], antioxidant, anti-inflammation [66], anti-cancer [67], and other properties [68]. Clinical trials employing two ITCs, sulforaphane (SFN; 1-isothiocyanato-4-(methylsulfinyl)butane) and phenethyl isothiocyanate (PEITC; 2-isothiocyanatoethylbenzene) to treat various diseases from cancer to autism [69]. More than 100 ITCs have been reported in a variety of plants, and ITCs often present in the diet and have anti-tumor effects mainly include allyl-isothiocyanate (AITC), benzylisothiocyanate (BITC), PEITC, and SFN [70]. ITCs can permanently bind some important biomolecules, such as cytoskeletal proteins, transcription factors, as well as heat shock proteins, while they also inhibit histone deacetylases, affecting the epigenetic regulation of gene expression [71].

ITCs, which are contained in the *M. oleifera* body [24], are formed by the glycosylated precursor GLs via myrosinase (thioglucoside glucohydrolase), which is an enzyme activated during plant tissue damage or digestion. Myrosinase cleaves thio-linked glucose in GLs, leaving a rapid rearrangement to form four active and stable ITCs [72]. Among them, 4-(α-L-rhamnosyloxy)benzyl isothiocyanate and 4-(4′-*O*-acetyl-α-L-rhamnosyloxy)benzyl isothiocyanate are the most abundant ITCs formed by GLs, usually accounting for more than 95% of the total ITCs. 4-(2′-*O*-acetyl-α-L-rhamnosyloxy)benzyl isothiocyanate and 4-(3′-*O*-acetyl-α-L-rhamnosyloxy)benzyl isothiocyanate are only formed from microscopic amount of GLs precursors [73]. The structures of these ITCs are shown in Figure 2. Among them, 4- (α-L-rhamnosyloxy)benzyl isothiocyanate also known as MIC-1, ITC-1, RBITC, and GMG-ITC. Huang et al. [37] extracted 200 g of seeds with MeOH/H_2_O and purified 230 mg of 4- (α-L-rhamnosyloxy)benzyl isothiocyanate.

### 3.2. The Extraction Methods of MIC-1 from M. oleifera

ITCs are contained in all parts of the *M. oleifera* body, but it is higher in *M. oleifera* seeds than in any other tissue [24]. GLs and ITCs are usually extracted with methanol or ethanol, and the specific manner as well as the extraction effect, are described by Lopez-Rodriguez et al. [23]. At present, the more mature extraction methods of MIC-1 in *M. oleifera* is established by Jaja-Chimedza et al. [74] and Brunelli et al. [75], other researchers obtained the experimental materials needed in their own experiments by subtly adjusting some steps in these two methods. The resulting MIC-1 produced a significant anti-cancer, antioxidant, and anti-inflammation activity (Table 2). In addition, some researchers have explored the way MIC-1 is extracted in *M. oleifera* and conducted relevant studies, as shown in Table 2.

## 4. In Vivo and In Vitro Studies of MIC-1 from *M. oleifera*

ITCs present in *M. oleifera* are a class of substances with multiple biological activities. Many researchers have designed a series of experiments to study their specific biological functions, in order to provide ideas for future relevant clinical trials. Treatment of mice with different concentrations of ITCs revealed that ITCs mainly involving abnormal changes in the digestive (gastrointestinal tract, liver), immune (thymus, spleen, WBC, lymphocytes) and (male) reproductive systems (testicular germinal cells, epididymides) at higher concentrations. It is also speculated from these results that MIC-1 appears to be less toxic than other ITCs [91].

### 4.1. Anti-Cancer

MIC-1 was found to have the ability to inhibit cancer cell proliferation, promote apoptosis of cancer cells, and inhibit metastasis. At the same time, after treatment with MIC-1 for HepG2, Caco-2, and HEK293, cell viability determination using MTT revealed that MIC-1 exhibited selective cytotoxic and apoptotic activity in human and non-cancer cells [92]. Besides, MIC-1 can downregulate some signaling pathways associated with cancer cell proliferation to inhibit the development of cancer cells [93].

#### 4.1.1. Neuroblastoma

Neuroblastoma (NBL) is the third-most common childhood malignancy following leukemia and brain tumors [94], and it is also the most common cancer type used to study the anti-cancer mechanism of MIC-1. Using SH-SY5Y human NBL cell line, researchers found that MIC-1 can inhibit the proliferation of malignant cell lines by activating apoptosis or programmed cell death. The phosphatidylinositol 3-kinase (PI3K)/ protein kinase B (Akt)/ mammalian target of rapamycin (mTOR) pathway is one of the most potent pro-survival pathways involved in the progression of NBL and is also associated with a poor prognosis [95,96]. Giacoppo, et al. [97] found that MIC-1 complexed with α-cyclodextrin can downregulate the levels of p-PI3K, p-Akt, and p-mTOR associated with this signaling pathway, thereby downregulating the signaling pathway and inhibiting SH-SY5Y cell survival. At the same time, MIC-1 can also inhibit the mitogen-activated protein kinase (MAPK) pathway, triggered by PI3K/Akt/mTOR signaling, which plays a key role in regulating many cellular functions, including survival, proliferation, and apoptosis in different cell types. In addition, MIC-1 can also upregulate p53 and p21 expression and promote caspase 3 cleavage, thus promoting apoptosis in SH-SY5Y cells. The study by Cirmi et al. [85] shown that MIC-1 can alter the normal progression of cells during the cell cycle, increase the number of cells in the G2 and S phases, reduce the number of cells in the G1 phase, and inhibit the nuclear translocation of NF-κB. Moreover, Jaafaru et al. [86] further found that MIC-1, even with oxidative damage by hydrogen peroxide, was able to preserve the membrane and internal structural integrity of differentiated neurons, suggesting its ability to protect neurons from oxidative stress degeneration. Thus it can be seen, targeting NBL deserves expansion to gain more evidence of how the compound provides these effects and promotes apoptosis, and the actual regulatory mechanistic pathways involved in the process.

#### 4.1.2. Astrocytomas

Astrocytomas are the most aggressive primary brain tumor, and are also the most common and malignant primary brain tumors in adults [98]. Human astrocytoma grade IV CCF-STTG1 cells by Rajan et al. [87] found that MIC-1 effectively induced apoptosis through activation of p53 and Bax as well as inhibition of Bcl-2. At the same time, it can also induce oxidative stress-mediated apoptosis by modulating the expression of the oxidative stress-associated Nrf2 transcription factor and its upstream regulator casein kinase 2 alpha. Specific signaling pathways that regulate apoptosis require further investigation.

#### 4.1.3. Hepatocarcinoma

Apoptosis is controlled by two pathways: the extrinsic pathway is activated by death ligands and their death receptor interactions, resulting in activation of initiator caspase 8 and effector caspase 3, and the intrinsic pathway regulated by initiator caspases 2 and 9, which in turn, activate effector caspase 3 [99]. Using MIC-1 and avenanthramide 2f (AVN 2f) to treat Hep3B hepatocarcinoma cells, Antonini et al. [100] found that MIC-1 and AVN 2f cocktail significantly inhibited the proliferation of Hep3B by increasing caspases 2, 8, 9, and 3 activity. Extrinsic apoptosis is apoptosis induced by AVN 2f -mediated activation of caspase 8, while the intrinsic apoptosis pathway is triggered by MIC-1-induced increased intracellular ROS levels, MIC-1-mediated activation of caspases 2 and 9, and MIC-1-mediated downregulation of the pro-survival gene BIRC5. The results suggest that the combination of MIC-1+AVN 2f may be an effective chemoprophylaxis cocktail against the development of liver cancer.

#### 4.1.4. Skin Carcinoma

TPA is an accelerator of cancer development and induces inflammation, oxidative imbalance, and excessive cell proliferation. It can also promote the development of skin cancer [101]. Wang et al. [102] studied mouse epidermal JB6 cells induced by the tumor promoter TPA, found that MIC-1 treatment could alter gene expression changes induced by TPA. Seventy-six pathways activated by TPA, were inhibited by MIC-1; and nine signaling pathways inhibited by TPA, were activated by MIC-1. While most of these signaling pathways are inflammatory responses, cancer, and oxidative stress-related pathways such as involvement in NF-κB signaling, IL-1 signaling, LPS/IL-1-mediated inhibition of RXR function, PTEN signaling, p53 signaling, and Nrf2-mediated oxidative stress responses. At the same time, MIC-1 can also affect the methylation of the gene through epigenetic modifications, which reduces CpG methylation in the promoter region more greatly than that in other regions. Thus, it appears that MIC-1 is chemoprophylactic on TPA-induced tumor/tumorigenic transformation in mouse epidermal JB6 cells, while MIC-1 can also relieve inflammation and oxidative stress induced by TPA. A potential set of transcriptome and epigenomic biomarkers was observed in the course of the study by Wang et al. These findings provide new insights into how epigenetic modifications affect the progression of skin carcinogenesis and the prophylactic role of MIC-1.

Thus, it is seen that MIC-1 not only inhibits the development of cancer by inhibiting inflammation and proliferation related factors, but also promotes the expression of genes related to apoptosis. However, relevant animal and clinical trials are lacking in the study, and the cancer cell types involved need to be further extended. Figure 3 shows the signaling pathways involved in MIC-1 regulation in cancer cells as well as the expression of related genes. At the same time, there was the finding that MIC-1 degrades rapidly to several water-soluble products via a pseudo-first-order kinetics, in a recent study by Lu et al. [103]. Therefore, although MIC-1 is highly cytotoxic for cancer cells, its degradation product is very weak or even without such activity, suggesting that other isothiocyanate components in ITCs are also important for their cancer chemoprevention.

### 4.2. Anti-Inflammation

A large number of studies in recent years have reported on the anti-inflammatory effects of ITCs. Lipopolysaccharide (LPS) is a powerful inducer of inflammation, endotoxemia, and sepsis in cells and animals [104]. In an LPS mouse model, it was found that oral administration of chemically stable MIC-1 (80 mg/kg) significantly reduced the expression of inflammatory markers in liver, kidney, spleen, and colon and reduced spleen weight. Meanwhile, it can also activate Nrf2 and inhibit NF-κB signaling, reduce ROS in the cytosol, enabling reduced mitochondrial peroxide content and restoration of mitochondrial membrane potential [105]. NO is an important inflammatory mediator produced by nitric oxide synthase, and MIC-1 treatment can also reduce NO production [74]. The results from the studies suggest the possible link between MIC-1 and anti-inflammation.

### 4.3. Anti-Chronic Diseases

#### 4.3.1. Anti-Diabetic

It is known that ulcerative colitis (UC) is an inflammatory bowel disease (IBD), a chronic intestinal disease characterized by recurrent inflammation in the gastrointestinal tract. Drug interventions are one of the key approaches for UC treatment. Using dextran sulfate sodium-induced acute UC mouse model, Kim et al. [106] found that MIC-1 decreases the secretion of colonic pro-inflammatory biomarkers and fecal lipocalin-2 levels, downregulates the colonic expression of pro-inflammatory cytokines and iNOS, and elevates the colonic expression of tight-junction proteins and Nrf2-mediated phase II detoxifying enzymes. MIC-1 improves the pathological events associated with acute and chronic UC, and the anti-inflammatory and antioxidant activity of MIC-1 may be related to Nrf2-mediated anti-inflammatory/antioxidant signaling. This study supports the therapeutic potential for MIC-1 prevention and treatment of UC, but should be further explored in human studies. ITCs also affects the onset of diabetes. Waterman, et al. [77] using MIC-1 to treat Type-2 diabetes mellitus (T2DM) rats, found that MIC-1 can delay the onset of diabetes in the T2DM rat model to a greater extent than moderate caloric restriction and it did not affect body weight or food intake.

#### 4.3.2. Anti-Obesity

Jaja-Chimedza et al. [25] suggested that the extraction of *M. oleifera* seeds significantly improved metabolic health in diet-induced obesity and insulin-resistant mice, most likely to be an antimicrobial effect of MIC-1. MIC-1 inhibits the gut microbiome, which may reduce metabolic endotoxemia and improve overall metabolic health. At the same time, in vitro studies found that MIC-1 can indeed downregulate genes associated with obesity, further confirming the efficacy with MIC-1 [76]. These results support its documented traditional uses and a bioactive role of MIC-1 and suggest the potential efficacy for *M. oleifera* supplementation for diabetes management in populations.

#### 4.3.3. Anti-MS

Multiple sclerosis (MS) is a chronic inflammatory disease, attacking the central nervous system, leads to demyelination, oligodendrocyte loss, glial scar formation, and subsequent degeneration of axonal and neuronal damage [107]. After treating the C57Bl/6 male mice with MIC-1, Galuppo et al. [108] found that GMG-ITC treatment is able to alleviate the inflammatory cascade behind the processes leading to severe MS. This suggests that the compound can be useful as a drug for the treatment or prevention of MS.

At the same time, researchers also found that MIC-1 can relieve neurologic pain by inhibiting inflammatory pathways and blocking voltage-gated ion channels in murine model of MS. These show that ITCs in *M. oleifera*, used clinically in the treatment of diseases, have great possibilities. However, the specific mechanisms involved, and the relevant precautions, also need to be determined through more clinical trials.

## 5. Conclusions

In conclusion, MIC-1 in *M. oleifera* has high safety and good physiological pharmacological activity and the cellular inhibitory effect of *M. oleifera* flowers, leaves, seeds, fruit, and bark shown in recent studies reflects the possibility of *M. oleifera* as a potential source of solutions to the current challenges faced in treating cancer. The soluble extract has a highly significant antitumor activity, likely due to the synergy of MIC-1 with other effective compounds in the extract that may enhance its activity. However, in-depth studies at the cellular and molecular levels are critical to address the limitations currently present. According to research findings that *M. oleifera* has a significant inhibitory effect on human lung, breast, prostate, pancreatic, and colorectal cancer cells, further well-designed in vivo studies should be conducted in order to provide more comprehensive information of *M. oleifera* and to fully exploit its potentiality in the clinical settings.

## Figures and Tables

**Figure 1 molecules-26-07512-f001:**
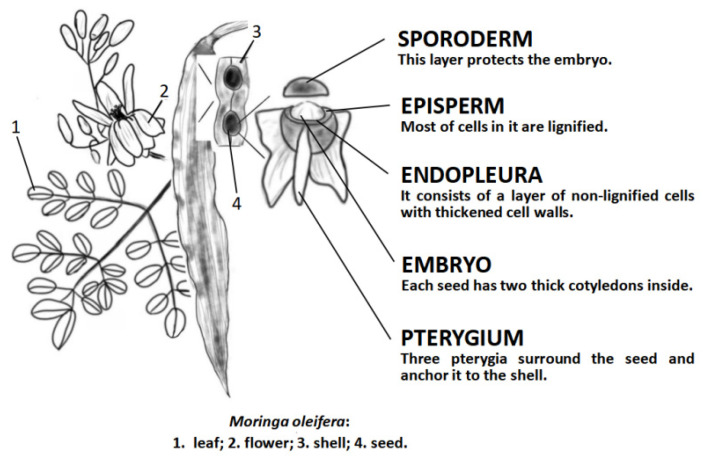
Schematics of *Moringa oleifera* representative body parts.

**Figure 2 molecules-26-07512-f002:**
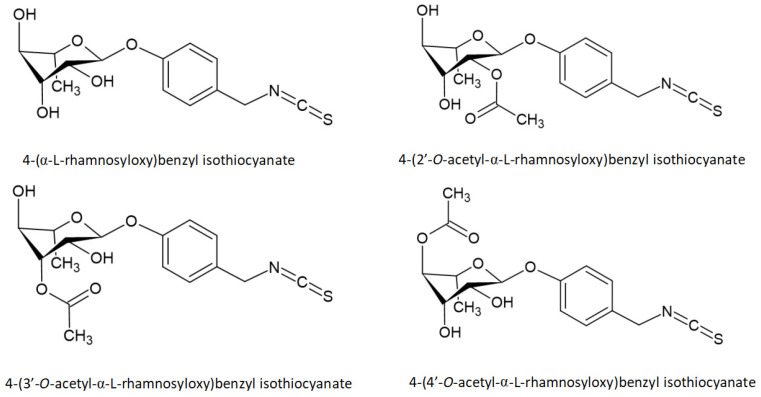
Chemical structures of moringa isothiocyanates.

**Figure 3 molecules-26-07512-f003:**
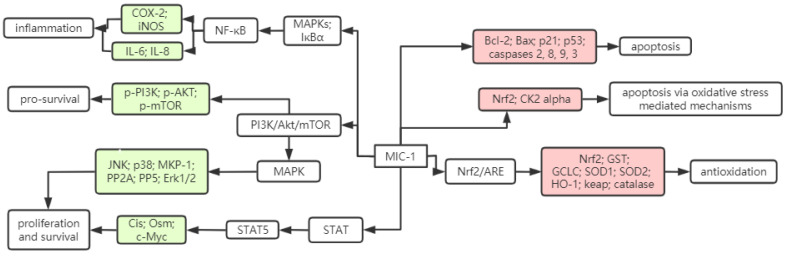
Proposed molecular target of MIC-1 as an anti-cancer agent. This figure is some of the signaling pathways that MIC-1 participates in cancer cells, as well as the proteins and genes regulated by these signaling pathways. Green boxes represent the downregulated genes or proteins, and red boxes represent the upregulated genes or proteins. COX-2: Cyclooxygenase-2; iNOS: Inducible nitric oxide synthase; IL-6: Interleukin-6; IL-8: Interleukin-8; JNK: c-Jun NH2-terminal kinase; p38: p38 Mitogen-activated protein kinase; MKP-1: Mitogen-activated protein kinase phosphatase-1; PP2A: Protein phosphatase 2A; Erk1/2: Extracelluar signal-regulated kinase 1/2; p-PI3K: Phosphorylated phosphatidylinositol 3-kinase; p-AKT: Phosphorylated protein kinase B; p-mTOR: Phosphorylated mammalian target of rapamycin; Cis: Cytokine-inducible SH2-containing protein; Osm: Oncostatin M; c-Myc: MYC proto-oncogene; Bcl-2: B-cell lymphoma 2; Bax: BCL2-associated X protein; p21: Cyclin-dependent kinase inhibitor 1; p53: Tumor suppressor p53; Nrf2: Nuclear factor erythroid-2-related factor 2; CK2: Casein kinase 2; GST: Glutathione S-transferase; GCLC: glutamate-cysteine ligase catalytic subunit; SOD1: Superoxide dismutase 1; SOD2: Superoxide dismutase 2; HO-1: Heme oxygenase 1; STAT: Signal transducer and activator of transcription.

**Table 1 molecules-26-07512-t001:** Major bioactive compound in different body parts of *M. oleifera*.

Location	Bioactive Compounds	Reference
Seed	glycosidic benzylamines; niazimicin; isothiocyanates; phenolics; glucosinolates	[35,36,37,38]
Leaf	phytol; flavonoids; phenolics; β-carotene; lycopene; vicenin-2; quinic acid; octadecanoic acid; hexadecanoic acid (palmitic acid); α-tocopherol (vitamin-E); ɣ-sitosterol	[32,39,40,41,42]
Flower	β-sitosterol; flavonoids; anthocyanin	[41]
Root	nasimizinol; oleic acid; *N*-benzyl-*N*-(7-cyanato heptanamide; *N*-benzyl-*N*-(1-chlorononyl) amide; bis [3-benzyl prop-2-ene]-1-one; *N*, *N*-dibenzyl-2-ene pent-1,5-diamide	[43]
Shell	3,5,6-trihydroxy-2-(2,3,4,5,6-pentahydroxyphenyl)-4H-chromen-4-one; β-sitosterol-3-*O*-glucoside; 2,3,4-trihydroxybenzaldehyde; stigmasterol	[44]
Bark	epiglobulol; flavonoids; anthocyanin	[41]

**Table 2 molecules-26-07512-t002:** The extraction methods and functions of MIC-1.

Location	Function	Extraction Method	Reference
Seed	anti-inflammation; anti-diabetic; counteracting ulcerative colitis	Jaja-Chimedza et al.	[25,74,76,77]
Seed	anti-cancer; counteracting neurodegeneration; anti-oxidation; counteracting neuropathic pain; anti-inflammation; antibiotics; counteracting amyotrophic lateral sclerosis	Brunelli et al.	[75,78,79,80,81,82,83,84,85,86,87,88]
Leaf	anti-inflammation; anti-obesity, anti-diabetic	Waterman et al.	[72,89]
Leaf	anti-inflammation	Fahey et al.	[90]

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
