# Peer review of "Anti-Cancer and Medicinal Potentials of Moringa Isothiocyanate"

_molecules, 2021, doi:10.3390/molecules26247512_

Round 1

Reviewer 1 Report

The manuscript is very interesting and well written and contains a lot of important information 

The manuscript can be accepted after minor revision for the grammar and some little typing mistakes. Also, It is better to add the doi for the references list to be easy to follow

Author Response

Reviewer: 1

Comments to the Author

The manuscript is very interesting and well written and contains a lot of important information. The manuscript can be accepted after minor revision for grammar and some little typing mistakes. Also, It is better to add the doi for the references list to be easy to follow.

Our response: Thank you very much for referee 1’s comments and constructive suggestions. We have addressed all the concerns by revising the grammar and typing mistakes, and adding the doi for the references list.

Overall changes in the revised manuscript

For the text, it has been revised extensively according to the comments from all reviewers. New references have been added and irrelevant references removed, and cited accordingly in all sections. All the possible English misusage, unfriendly mode of writing, and possible typos have been corrected. All changes have been tracked and indicated on the right of each page.

For the figures, the original Figures 1, 2 were revised, and in figure 1 we added some details in the revised manuscript.

For the tables, we have added one table about major bioactive compound in different body parts of M. oleifera.

Other major changes in the revised manuscript:

- The number of references, from a total of 90 in the original submission, after adding/removing new and old references, the total number is 108 now in the revised manuscript.

We hope that you will find our revised manuscript with satisfaction and thank you for your time for reviewing it.

Reviewer 2 Report

This is an interesting review paper that explores the anticancer potential of Moringa oleifera MIC-1. This manuscript would be definitely a useful addition to the literature and hopefully will encourage further investigation. The data and writing are well done; however, some minor issues should be addressed before presenting a potentially acceptable version of the manuscript for reconsideration.

  • Overall the paper is well written however there are a few places where the wording is a bit clumsy or prepositions are missing. These don’t detract from the meaning of the text but careful proofread could address these issues and improve the flow of the text.
  • Line number 86 should be reframed, “this shows…….MIC-1.
  • In section 2, there are some other studies which explored the anticancer potential of Moringa oleifera, try to cite them too.
  • Is there any clinical study which explores the anticancer potential of MIC-1?
  • Please explain the significance of extraction method used for the isolation of MIC-1. Is there any need of extraction method in this review?
  • Authors have also explored the pharmacological properties of MIC-1, however they focused their review on anticancer properties of MIC-1. Please discuss.
  • Authors are suggested to improve the quality of figures.

Author Response

Reviewer: 2

Comments to the Author

This is an interesting review paper that explores the anticancer potential of Moringa oleifera MIC-1. This manuscript would be definitely a useful addition to the literature and hopefully will encourage further investigation. The data and writing are well done; however, some minor issues should be addressed before presenting a potentially acceptable version of the manuscript for reconsideration. Overall the paper is well written however there are a few places where the wording is a bit clumsy or prepositions are missing. These don’t detract from the meaning of the text but careful proofread could address these issues and improve the flow of the text. Line number 86 should be reframed, “this shows………MIC-1”. In section 2, there are some other studies which explored the anticancer potential of Moringa oleifera, try to cite them too. Is there any clinical study which explores the anticancer potential of MIC-1? Please explain the significance of extraction method used for the isolation of MIC-1. Is there any need of extraction method in this review? Authors have also explored the pharmacological properties of MIC-1, however they focused their review on anticancer properties of MIC-1. Please discuss. Authors are suggested to improve the quality of figures.

Our response: Thank you very much for referee 2’s comments and constructive suggestions. We have addressed all the concerns and listed point by point as below.

1) There are a few places where the wording is a bit clumsy or prepositions are missing. These don’t detract from the meaning of the text but careful proofread could address these issues and improve the flow of the text. Line number 86 should be reframed, “this shows………MIC-1”.

Our response: We apologize for this. In the revised manuscript, we have performed thorough language editing and carefully checked the entire manuscript staring with the title.

2) In section 2, there are some other studies which explored the anticancer potential of Moringa oleifera, try to cite them too.

Our response: We agree with you on this point. In the revised manuscript, we have added some of studies about the anticancer potential of Moringa oleifera. You can access these paper through PubMed. Their PubMed ID as below: 32764438; 30945951; 33302731; 32550010; 28573245; 26107222.

3) Is there any clinical study which explores the anticancer potential of MIC-1?

Our response: Thank you for your suggestion and we agree with you that it would be nice if we can include some data with respect to the clinical trials. However, it may be still impossible to give some perspective comments due to the fact that there is no information of the results of these clinical trials yet that have been performed on MIC-1.

4) Please explain the significance of extraction method used for the isolation of MIC-1. Is there any need of extraction method in this review?

Our response: We agree with you on this point. After careful thought, we found that there was indeed too much discussion about the extraction part, and that the MIC-1 can be purchased directly in the company. We therefore removed the specific process of the extraction. However, because in the laboratory, the MIC-1 extracted from Moringa oleifera may contain some other compounds mixed in them, and the interaction of these compounds on cancer cells is irreplaceable by MIC-1 alone, some new anticancer mechanisms and anticancer drugs may be found. Therefore, we still retained some of the extraction methods, so that future researchers can learn from the extraction methods, but we did not elaborate on them in detail.

5) Authors have also explored the pharmacological properties of MIC-1, however they focused their review on anticancer properties of MIC-1. Please discuss.

Our response: Thank you for your suggestion. After thinking, we changed the title and abstract section of the article to make it more reasonable to discuss the pharmacological properties of MIC-1. The reason for our discussion of the medicinal value of MIC-1 is that cancer is also a chronic disease, and from current studies we understand that the effects of MIC-1 on these diseases mostly occur above the same mechanisms. Discussion of these diseases can give researchers a better understanding of the treatment mechanism of MIC-1 and facilitate future anti-cancer research.

6) Authors are suggested to improve the quality of figures.

Our response: Thank you for your suggestions. We have redrawn all the figures and added more details.

Overall changes in the revised manuscript

For the text, it has been revised extensively according to the comments from all reviewers. New references have been added and irrelevant references removed, and cited accordingly in all sections. All the possible English misusage, unfriendly mode of writing, and possible typos have been corrected. All changes have been tracked and indicated on the right of each page.

For the figures, the original Figures 1, 2 were revised, and in figure 1 we added some details in the revised manuscript.

For the tables, we have added one table about major bioactive compound in different body parts of M. oleifera.

Other major changes in the revised manuscript:

- The number of references, from a total of 90 in the original submission, after adding/removing new and old references, the total number is 108 now in the revised manuscript.

We hope that you will find our revised manuscript with satisfaction and thank you for your time for reviewing it.

Reviewer 3 Report

This manuscript describes the effect of  Isothiocyanates (ITCs) from M. Oleifera on cancer cells.

From my perspective, such a subject deserves the scientific public's attention, I recommend a major revision before the present manuscript  can be accepted for publication.

Although some data have original aspects to some extent, this work is preliminary and descriptive without providing a full chemical marker compounds of M. Oleifera and their relevant mechanistic insights.

I suggest changing the title. What does MIC-1 represent? It  is 4-[(α-L rhamnosyloxy) benzyl] isothiocyanate ? Can be confused with macrophage inhibitory cytokine (abbreviated also MIC-1)

I think that the manuscript can be improved first of all from the point of view of clarity, scientific rigorousness and language

Please highlight in a better way what are the motives to perform your study, way you would like  to present only the anticancer activity of the isocyanates from M.oleifera, although there are also other compounds with anticancer properties?

On point 4, the  authors presented also other properties than anti-cancer. It seems to me that these biological activities do not fall within the proposed topic, "isothiacyanates with anticancer properties". I consider that the paper must be majorly modified. Or the authors present the anticancer properties of M.oleifera, also for other biologic compounds from  this plant, or they will present all the biological properties of this plant, and then points 4.2-4.3 may be also presented.

Author Response

Reviewer: 3

Comments to the Author

This manuscript describes the effect of Isothiocyanates (ITCs) from M. oleifera on cancer cells. From my perspective, such a subject deserves the scientific public’s attention, I recommend a major revision before the present manuscript can be accepted for publication. Although some data have original aspects to some extent, this work is preliminary and descriptive without providing a full chemical marker compounds of M. oleifera and their relevant mechanistic insights. I suggest changing the title. What does MIC-1 represent? It is 4-[(a-L rhamnosyloxy) benzy] isothiocyanate ? Can be confused with macrophage inhibitory cytokine (abbreviated also MIC-1). I think that the manuscript can be improved first of all from the point of view of clarity, scientific rigorousness and language. Please highlight in a better way what are the motives to perform your study, way you would like to present only the anticancer activity of the isothiacyanates from M. oleifera, although there are also other compounds with anticancer properties? On point 4, the authors presented also other properties than anti-cancer. It seems to me that these biological activities do not fall within the proposed topic,”isothiacyanates with anticancer properties”. I consider that the paper must be majorly modified. Or the authors present the anticancer properties of M.oleifera, also for other biologic compounds from this plant, or they will present all the biological properties of this plant, and then points 4.2-4.3 may be also presented.

Our response: Thank you very much for referee 3’s comments and constructive suggestions. We have addressed all the concerns and listed point by point as below.

1) Although some data have original aspects to some extent, this work is preliminary and descriptive without providing a full chemical marker compounds of M. oleifera and their relevant mechanistic insights.

Our response: We agree with you on this point. In the revised manuscript, we have added the discussion about bioactive compounds form M. oleifera in section 2. However, because the article mainly wants to show the cancer research of MIC-1, the excessive discussion of other substances in M. oleifera may deviate from the subject matter of the article. Thank you for your understanding.

2) I suggest changing the title. What does MIC-1 represent? It is 4-[(a-L rhamnosyloxy) benzy] isothiocyanate ? Can be confused with macrophage inhibitory cytokine (abbreviated also MIC-1).

Our response: Thank you for your suggestion. Considering that the MIC-1 you mentioned is easily confused with macrophage inhibitory cytokine, we changed the title to Anti-Cancer and Medicinal Potentials of Moringa Isothiocyanate.

3) I think that the manuscript can be improved first of all from the point of view of clarity, scientific rigorousness and language.

Our response: We apologize for this. In the revised manuscript, we have performed thorough language editing and carefully checked the entire manuscript staring with the title.

4) Please highlight in a better way what are the motives to perform your study, way you would like to present only the anticancer activity of the isothiacyanates from oleifera, although there are also other compounds with anticancer properties?

Our response: We agree with you on this point. In the revised manuscript, We added a discussion of MIC-1 in the introduction section, highlighting our motivation to launch the study.

5) On point 4, the authors presented also other properties than anti-cancer. It seems to me that these biological activities do not fall within the proposed topic,”isothiacyanates with anticancer properties”. I consider that the paper must be majorly modified. Or the authors present the anticancer properties of oleifera, also for other biologic compounds from this plant, or they will present all the biological properties of this plant, and then points 4.2-4.3 may be also presented.

Our response: Thank you for your suggestion. After thinking, we changed the title and abstract section of the article to make it more reasonable to discuss the pharmacological properties of MIC-1. The reason for our discussion of the medicinal value of MIC-1 is that cancer is also a chronic disease, and from current studies we understand that the effects of MIC-1 on these diseases mostly occur about the same mechanisms. Discussion of these diseases can give researchers a better understanding of the treatment mechanism of MIC-1 and facilitate future anti-cancer research. We agree with you on this point. In the revised manuscript, we have added the discussion about bioactive compounds form M. oleifera. However, because the article mainly wants to show the cancer research of MIC-1, the excessive discussion of other substances in M. oleifera might not match the subject matter of the article.

Overall changes in the revised manuscript

For the text, it has been revised extensively according to the comments from all reviewers. New references have been added and irrelevant references removed, and cited accordingly in all sections. All the possible English misusage, unfriendly mode of writing, and possible typos have been corrected. All changes have been tracked and indicated on the right of each page.

For the figures, the original Figures 1, 2 were revised, and in figure 1 we added some details in the revised manuscript.

For the tables, we have added one table about major bioactive compound in different body parts of M. oleifera.

Other major changes in the revised manuscript:

- The number of references, from a total of 90 in the original submission, after adding/removing new and old references, the total number is 108 now in the revised manuscript.

We hope that you will find our revised manuscript with satisfaction and thank you for your time for reviewing it.

Round 2

Reviewer 3 Report

The authors have improved the manuscript and believe that it can be published. I would also have a suggestion regarding figure 2.Chemical structures of isothiocynates, which I don't think looks too good, the symbols of the atoms being too small compared to the valences, which also have various dimensions. Is it written in chemdraw?

Author Response

Review Report (Reviewer 3)

The authors have improved the manuscript and believe that it can be published. I would also have a suggestion regarding figure 2.Chemical structures of isothiocynates, which I don't think looks too good, the symbols of the atoms being too small compared to the valences, which also have various dimensions. Is it written in chemdraw?

Our response: We apologize for that and we agree with your suggestion. The chemical structures have now been rewritten by using Chemsketch, which should be looking nice and clear now.